# Dynamic similarity and the peculiar allometry of maximum running speed

**David Labonte** [1] ✉, **Peter J. Bishop**[2,3], **Taylor J. M. Dick** [4] & **Christofer J. Clemente** [4,5]

Animal performance fundamentally influences behaviour, ecology, and evolution. It typically varies monotonously with size. A notable exception is maximum running speed; the fastest animals are of intermediate size. Here we show that this peculiar allometry results from the competition between two musculoskeletal constraints: the kinetic energy capacity, which dominates in small animals, and the work capacity, which reigns supreme in large animals. The ratio of both capacities defines the physiological similarity index Γ, a dimensionless number akin to the Reynolds number in fluid mechanics. The scaling of Γ indicates a transition from a dominance of muscle forces to a dominance of inertial forces as animals grow in size; its magnitude defines conditions of "dynamic similarity" that enable comparison and estimates of locomotor performance across extant and extinct animals; and the physical parameters that define it highlight opportunities for adaptations in musculoskeletal "design" that depart from the eternal null hypothesis of geometric similarity. The physiological similarity index challenges the Froude number as prevailing dynamic similarity condition, reveals that the differential growth of muscle and weight forces central to classic scaling theory is of secondary importance for the majority of terrestrial animals, and suggests avenues for comparative analyses of locomotor systems.

The variation of locomotor performance with animal size is of substantial ecological and evolutionary importance, and has thus long been a topic of interest in comparative animal physiology and biomechanics[1–6]. Empirical data show that maximum running speed increases up to a critical body mass and then decreases—the fastest runners are of intermediate size (Fig. 1). This pattern is a noteworthy outlier among scaling relationships, which typically are monotonous and satisfactorily described by simple power laws[7,8]; it has consequently attracted persistent attention[6,9–19]. What explains the peculiar allometry of maximum running speed?

A common starting point for the mechanistic analysis of scaling relationships are similarity arguments that lead to characteristic dimensionless numbers[20,21]. An increment in speed $v$ of a body with

mass $m$ implies that work $W$ was done, so that a general dimensionless number for speed is:

$$\Pi \propto \frac{mv^2}{W} \tag{1}$$

Specific scaling predictions then depend on the origin of the work $W$, and each distinct origin defines a dimensionless index of 'dynamic similarity', $\Pi_i$. Animals that use the same source of work move with equal $\Pi_i$, and, in that sense, each $\Pi_i$ may be interpreted as a concrete hypothesis on the physical origin of motion, which can then be tested against empirical data. In walking, for example, the gravitational potential energy ($E_{pot}$) of the centre-of-mass (CoM) is

[1]Department of Bioengineering, Imperial College London, London, UK. [2]Museum of Comparative Zoology, Harvard University, Cambridge, MA, USA. [3]Geosciences Program, Queensland Museum, Brisbane, QLD, Australia. [4]School of Biomedical Sciences, University of Queensland, Brisbane, QLD, Australia. [5]School of Science and Engineering, University of the Sunshine Coast, Sippy Downs, QLD, Australia. ✉e-mail: d.labonte@imperial.ac.uk

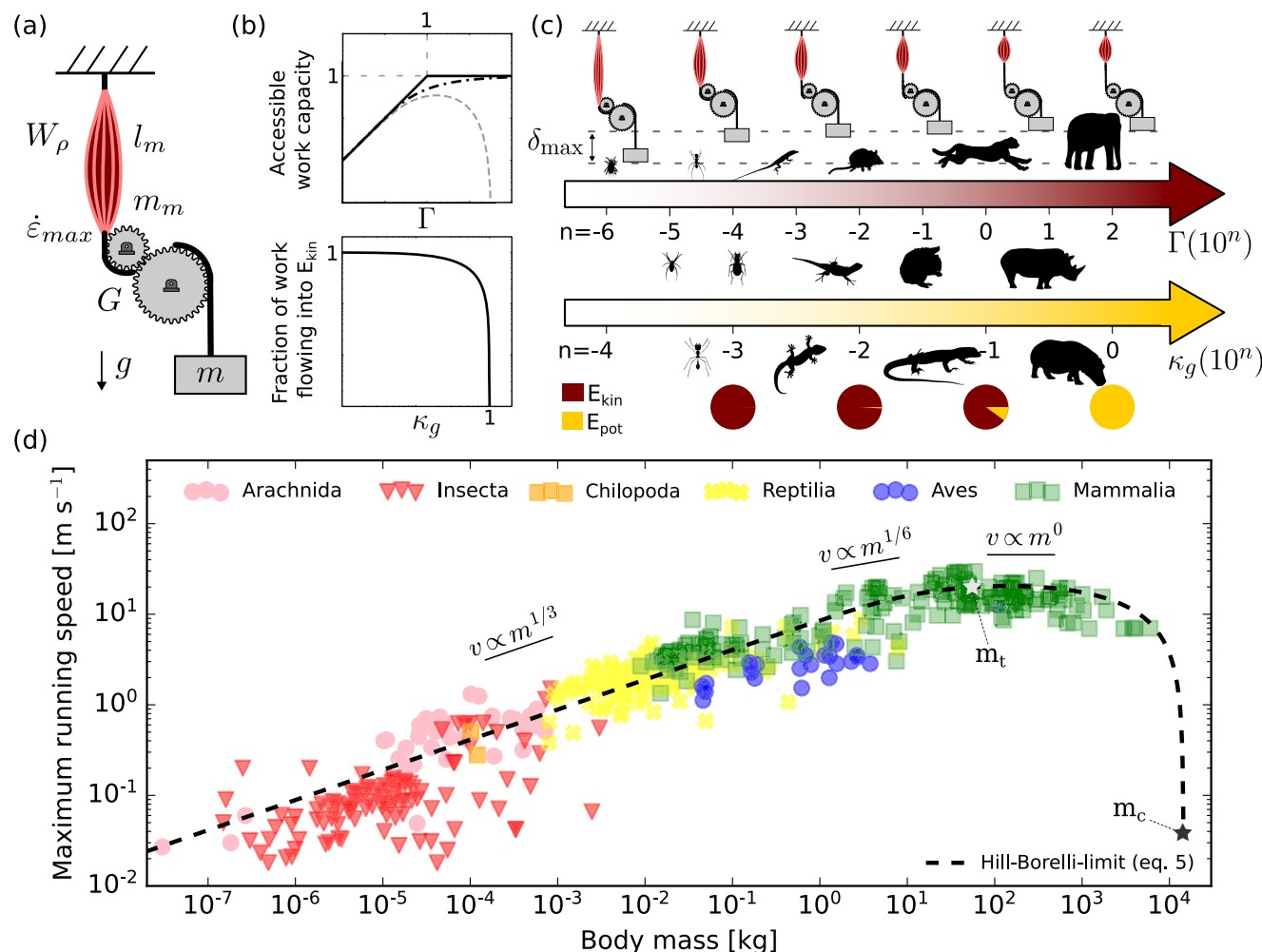

**Fig. 1 | Animals small and large move by using muscle as a motor, but the maximum running speed they can achieve varies non-monotonously with size: the fastest animals are of intermediate size. a** Schematic of a minimalistic physical model of a musculoskeletal system, defined by the muscle work density, $W_\rho$, the muscle fascicle length, $l_m$, the maximum muscle strain rate, $\dot{\varepsilon}_{max}$, the muscle mass, $m_m$, the gear ratio $G$ and the mass $m$ that is moved. Terrestrial locomotion also involves the gravitational acceleration $g$. **b** The performance space of this minimalistic system is fully characterised by two dimensionless numbers: the physiological similarity index, $\Gamma \sim m(l_m \dot{\varepsilon}_{max})^2 (W_\rho m_m)^{-1} G^{-2}$, and the reduced parasitic energy, $\kappa_g \sim mg(F_{max}G)^{-1}$. $\Gamma$ quantifies the competition between the kinetic energy and work capacity of muscle: for $\Gamma \leq 1$, the system can only deliver a fraction $\Gamma$ of its maximum work capacity, and for $\Gamma \geq 1$ it has access to its full work capacity (solid line). For a muscle force that is independent of muscle strain rate, the transition between these regimes is sharp and occurs at a body mass $m_t$ (see (**d**)); if the muscle has force-velocity properties, it is more gradual (dot-dashed line). $\kappa_g$ quantifies the fraction of muscle work which flows into kinetic vs gravitational potential energy. The energy demanded by gravity only becomes appreciable for large $\kappa_g$, eventually resulting in a sharp asymptote at a critical body mass, $m_c$, at which movement is no longer possible (grey dashed line, see (**d**)). **c** Both dimensionless numbers vary systematically with size for geometrically similar animals ($\Gamma \propto m^{2/3}$ and $\kappa_g \propto m^{1/3}$). As a consequence of the increase of $\Gamma$, larger animals have access to a larger fraction of their work capacity and are thus generally faster. However, due to the increase in $\kappa_g$, an increasingly larger share of this work has to pay for fluctuations in gravitational potential energy, eventually resulting in a reduction in speed (see (**b**)). **d** The combination of both effects results in the peculiar allometry of maximum running speed ($n = 633$); the black dashed line is a least-square fit of Eq. (5), leaving only a dimensionless scaling coefficient as free parameter (see text). $\Gamma$ thus emerges as a fundamental dimensionless number for musculoskeletal dynamics, which may be used and interpreted akin to the Reynolds number (see discussion). The three short solid lines illustrate asymptotic scaling relations defined by three alternative indices of 'dynamic similarity", $v_{Hi} \propto m^{1/3}$, $v_{Fr} \propto m^{1/6}$ and $v_{Bo} \propto v_{St} \propto m^0$ (see text). Source data for (**d**) are provided as a Source Data file.

cyclically exchanged for kinetic energy ($E_{kin}$), so that $W = mg\Delta L$, where $g$ is the gravitational acceleration, and $\Delta L$ is the characteristic fraction of the leg length over which the CoM drops as the animal 'falls' forward during each step. The associated dynamic similarity index is $\Pi_{Fr} = v^2(g\Delta L)^{-1} = Fr^2$ – the square of the Froude number, Fr, introduced to terrestrial locomotion in seminal work by Alexander[22]. The Froude number predicts $v_{Fr} \propto m^{1/6}$ for geometrically similar animals, and is the appropriate dynamic similarity index when $E_{pot}$ and $E_{kin}$ fluctuate out-of-phase during stance[23]. It consequently cannot be used to predict speed for bouncing gaits such as running or hopping, for $E_{pot}$ and $E_{kin}$ now fluctuate in-phase during stance instead[23]. This in-phase fluctuation resembles the motion of a mass on a spring,

which has invited the proposition that $W = E_{ela} \propto k\Delta L^2$; kinetic energy is traded with the elastic strain energy ($E_{ela}$), cyclically stored in and released from a conceptual spring with spring constant $k$, deformed by a maximal characteristic displacement $\Delta L$. The associated dynamic similarity index is $\Pi_{St} \propto mv^2(k\Delta L^2)^{-1} = St^2$ – the square of a Strouhal number, St, implicitly applied to bouncing gaits in seminal work by Blickhan[24]. For geometrically similar animals, the Strouhal number predicts $v_{St} \propto \sqrt{k/m}\Delta L \propto m^0 = $ constant, which only holds in a narrow body mass region (Fig. 1). Thus, neither of the two established dynamic similarity indices provides a complete account for the non-monotonous scaling of maximum running speed. Explaining the origin of this peculiar allometry with simple similarity

arguments in the spirit of previous pioneering work is the core aim of this study.

## Results

Perhaps remarkably, muscle—the ubiquitous animal motor—plays no or at most an implicit role in classic dynamic similarity theory. What constrains the speed in movements directly actuated by muscle? One limit has been suggested in some of the earliest work on animal biomechanics[1,2,25]: each unit mass of muscle $m_m$, the argument goes, can deliver no more than a characteristic amount of mass-specific work, $W_\rho$. Introducing the ratio $m_f$ between muscle mass ($m_m$) and driven mass ($m$), $m_f = m_m/m$, yields the similarity index $\Pi_{Bo} \propto v^2(W_\rho m_f)^{-1} \propto Bo^2$—proportional to the square of the Borelli-number, $Bo$[26]. Much like the Strouhal number, the Borelli-number predicts a mass-invariant speed for geometrically similar animals, $v_{Bo} \propto \sqrt{W_\rho m_f} \propto m^0 =$ constant. Consequently, the Borelli-number, too, does not provide a satisfactory explanation for the observed allometry of maximum running speed.

The shortcoming of the Borelli number is well appreciated in the literature, and a good many alternatives have been proposed [see e.g.[14,18,27–30]]. The most popular idea has been that a fixed power density of muscle defines a speed limit distinct from the Borelli-limit, a line of thought which goes back to seminal work by Bennet-Clark in the 1970s[27], and which has been followed and developed further by several others since [e.g.[14,18,28,31–34]]. We will discuss the concept of a power limit to speed in detail separately[35], and note briefly that a finite power density cannot be linked to speed using only the moved mass—any limit to speed implies an explicit constraint on work output, and a variation in muscle power is thus neither necessary nor sufficient to alter the maximum possible speed. It has been argued that large animals cannot utilise the full work capacity of their muscles because they are limited by a supposed maximum anaerobic capacity[17]. However, the implementation of this physiological argument is beset by physical and mathematical errors [see Supplementary Discussion 1 and ref.[19]]. Recently, and as an alternative to a supposed power limit, it was suggested that muscle may be prevented from delivering its maximum work capacity because it cannot contract faster than with a characteristic maximum strain rate, $\dot{\varepsilon}_{max}$[26]. Where motion is directly actuated by muscle, this muscle strain rate is coupled to the speed thus imparted, and muscle work output is consequently bound by its 'kinetic energy capacity', $W \propto m l_m^2 \dot{\varepsilon}_{max}^2 G^{-2}$, where $G$ is the gear ratio of the musculoskeletal system, and $l_m$ is a characteristic muscle fascicle length (Fig. 1a). The associated dynamic similarity index follows as $\Pi_{Hi} \propto (Gv)^2 (l_m \dot{\varepsilon}_{max})^{-2} = Hi^2$—the square of the Hill-number, $Hi$[26]. The constraint encoded by the Hill-number may appear reminiscent of the suggestion that muscle 'force-velocity effects' limit work output in smaller animals[30,36]; it is however qualitatively distinct, for even a muscle which generates a force that remains constant up to a critical shortening velocity is bound by it. Movement with equal Hill-numbers implies running speeds that scale as $v_{Hi} \propto l_m \propto m^{1/3}$ for geometrically similar animals[19,26], close to what is observed in lightweight animals [Fig. 1 and e.g. refs.[10,14,19,37]], but incorrect for heavy animals.

In isolation, neither the Borelli- nor the Hill-number account for the peculiar allometry of maximum running speed. This shortcoming can be resolved by recognising that both dynamic similarity indices define an absolute limit on muscle mechanical output—the realised speed increment can be no larger than whichever of the two predicted increments is smaller[26]. Because the Hill- and the Borelli-limit scale differently for geometrically similar animals, $v_{Hi} \propto m^{1/3}$ vs. $v_{bo} \propto m^0$, there exists a transition mass, $m_T$, at which the limiting speeds they predict are equal [a conceptually similar idea, based on a limiting power density, was first presented in ref.[28]. For further details, see ref.[26]]. A direct calculation on the basis of empirical data suggests

$m_t \approx 54$ kg (see Methods), a physical prediction firmly grounded in first principles, and situated between statistical estimates based on polynomial models [$m_t \approx 119$ kg, see ref.[6]], or linear breakpoint regressions [$m_t \approx 31$ kg, see ref.[38]]. Animals lighter than $m_t$ are Hill-limited and speed consequently increases with size in this regime; animals heavier than $m_t$ are Borelli-limited, and speed is now independent of size (Fig. 1a–c). This pattern begins to qualitatively resemble the peculiar allometry of maximum running speed (solid line in Fig. 1b), but two noteworthy discrepancies remain: running speed does not remain constant in heavy animals, but instead decreases; and the scaling of running speed in animals of intermediate body mass is in fact shallower than $v_{Hi} \propto m^{1/3}$[6,39].

To explain the decrease of speed in the heaviest terrestrial animals, we note that the work in Eq. (1) is strictly the work done by the net force, $F_{net}$. But muscle is practically never the sole determinant of the net force, and instead has to frequently act against 'parasitic' forces ($P$), so that $F_{net} = F_m G - P$. As a consequence, only some muscle work flows into kinetic energy; the rest is lost to parasitic energy instead [[26,29], and see Fig. 1b, c]. This work loss must be accounted for in Eq. (1):

$$\Pi \propto \frac{mv^2}{W_m - W_p} \propto \frac{mv^2}{W_m}\frac{1}{1-\kappa} \tag{2}$$

Here, $W_p$ is the work done by the parasitic force, and $\kappa = P F_{max}^{-1} G^{-1}$ is the reduced parasitic energy, a dimensionless number which characterises the extent to which muscle work is 'consumed' by external forces [$F_{max}$ is the maximum force muscle can exert. See Fig. 1b, c and ref.[26]]. For terrestrial locomotion, the key parasitic force is the gravitational force, $P_g = mg$, so that $\kappa_g = mg(F_{max}G)^{-1} \propto m^{1/3}$ for geometrically similar animals [losses to drag are negligible, see[19]]. Equation (2) leaves no room for ambiguity: the possible speed increment is zero for $\kappa_g = 1$. The driving force $F_{max}G$ is then equal to the opposing force $mg$, and the system is in static equilibrium—a force-limit to muscle-driven motion, which occurs at a critical mass $m_c = F_{max}Gg^{-1}$ (or about 15 t, see methods). En route to this asymptote, the speed drops gently at first but then with an increasing rate (grey dashed line in Fig. 1b), qualitatively and quantitatively consistent with the empirical data (Fig. 1d).

In search for the mechanistic underpinning of the shallower scaling in animals with a body mass close to but below $m_t$, we propose two non-mutually exclusive explanations. First, we note that there is robust evidence that the musculoskeletal gear ratio, $G$, breaks with geometric similarity in some animal groups—large quadrupedal mammals tend to have larger gear ratios than small quadrupedal mammals, $G \propto m^{0.16}$ [Fig. 2 and refs.[40–43]]. The Hill-number then predicts $v_{Hi} \propto l_m \dot{\varepsilon}_{max} G^{-1} \propto m^{1/3} m^{-0.16} \propto m^{0.17}$, in close agreement with scaling coefficients reported for mammals [$v \propto m^{0.17}$ and $v \propto m^{0.16}$, see refs.[6,10]]. Second, we make explicit the thus far implicit assumption that the muscle force is independent of muscle strain ($\varepsilon$), and strain rate ($\dot{\varepsilon}$). This simplification is common and convenient, but only yields first order approximations. A simple implementation of more realistic muscle properties assigns a linear force-strain rate relationship, $F_m(\dot{\varepsilon}) = F_{max}(1 - \dot{\varepsilon}\dot{\varepsilon}_{max}^{-1})$ [for more complex calculations, see e.g. refs.[19,26,36]], and assumes a maximum muscle strain so small that muscle effectively remains on the plateau of the force-strain curve, $F_m(\varepsilon) \approx F_{max}$[44–46]. Because the force–strain relationship carries no new physical parameters, its effect cannot be assessed via simple dimensional arguments; the maximum work output can however be evaluated directly via conservation of energy, which yields a maximum speed [see ref.[26], and SI]:

$$v_{Hi-Bo} = v_{Hi}\left[1 + W\left(-\exp\left(-1 - \frac{1}{2\Gamma}\right)\right)\right] \tag{3}$$

Here, W is the Lambert-W function. The speed predicted by Eq. (3) cannot exceed the Hill limit, but it can be smaller: the difference is determined by the magnitude of the 'physiological similarity index' $\Gamma$[26]:

$$\Gamma = \frac{\text{Bo}^2}{\text{Hi}^2} = \frac{1}{2}\left[\frac{l_m^2}{G^2}\right]\left[\frac{\dot{\varepsilon}_{max}^2}{W_{\rho,max}}\right]\left[m_f\right] = \frac{E_{max}}{W_{max}} \qquad (4)$$

$\Gamma$ is the squared ratio between the Borelli and the Hill number, and was already leaned on implicitly to calculate the transition mass, $m_t$—the mass for which the Hill- and Borelli-number are equal, i.e. $\Gamma = 1$. $\Gamma$ quantifies the extent to which muscle work output is limited by the maximum kinetic energy capacity, $E_{max}$, or by the work density, $W_{max}$ [see Fig. 1b, c and ref. 26, for a more extensive interpretation of $\Gamma$]. The Hill-Borelli number, Hi-Bo = v $v_{\text{Hi-Bo}}^{-1}$, reduces to the Hill- and the Borelli-number in the limit of small and large $\Gamma$, respectively; the effect of the force-strain rate relationship is that it smoothens the otherwise sharp transition between the Hill- and the Borelli-limit around $\Gamma = 1$ [see dot-dashed line in Fig. 1b and refs. 26,30,36]. As a consequence, the running speed of animals with a body mass close to but below $\Gamma = 1$ generally scales between $v_{\text{Hi}} \propto m^{1/3}$ and $v_{\text{Bo}} \propto$ constant—a manifestation of 'force-velocity-effects'[30,36] that is in robust agreement with empirical data (Fig. 1d).

Gravity demands a reduction in absolute speed for heavy animals in the Borelli-limit, and size-specific variations in gear ratio and force-velocity effects reduce the speed in animals of intermediate weight in the Hill-limit; these effects may also be assessed in combination. Equation (3) then gains two additional terms that account for the loss of work to gravitational potential energy, and for the truncation of the strain rate accessible to muscle [see Supplementary Note 1 and also refs. 26,47–49]:

$$v_{\text{Hi-Bo},g} = v_{\text{Hi}}\left\{(1-\kappa_g)\left[1 + W\left(-\exp\left(-1-\frac{1}{2\Gamma}\frac{1}{1-\kappa_g}\right)\right)\right]\right\} \qquad (5)$$

Equation (5) reduces to Eq. (3) for $\kappa_g \to 0$ (i.e. for small animals), and is the main result of this paper, for it combines all elements of the preceding stepwise analysis. It paints the following mechanistic picture of the peculiar allometry of maximum running speed: for small animals, the kinetic energy capacity of muscle far exceeds its work capacity, $\Gamma \ll 1$, and gravity only demands a small toll, $\kappa_G \ll 1$; dynamic similarity is defined in terms of approximately equal Hill-numbers, animals run with approximately equal ratios of muscle strain rate to gear ratio, and $v \propto m^{1/3}$. In practise, and for all but the lightest animals, the scaling is reduced further by positive allometry of the gear ratio, and by the work loss arising from the force-strain rate relationship[26,30,36]; animals move with approximately equal Hill-Borelli numbers, and the maximum achievable strain rate slowly decreases with body mass. As soon as animals exceed the transition mass $m > m_T$, the work capacity of muscle is larger than its the kinetic energy capacity, $\Gamma > 1$, so that speed remains approximately constant; animals now run with approximately equal Borelli-numbers, and thus approximately equal mass-specific work output. For even heavier animals, speed decreases at an ever-increasing rate, because an increasing fraction of the muscle work is consumed by the gravitational force, $\kappa_g \to 1$[26,29]. Eventually, at $\kappa_g(m_c) = 1$, all animals can hope to achieve is to balance the gravitational force, no muscle-driven vertical movement is possible at all, and the speed plummets to zero.

In further support of this mechanical analysis, we next demonstrate that its qualitative scaling predictions can be rendered semi-quantitative. To this end, we once more draw from empirical estimates (Table 1), and directly predict the allometry of absolute maximum running speed. Because our analysis is based on dimensional considerations, it does not reveal numerical (non-dimensional) prefactors. We thus conduct a least-squares fit of Eq. (5) in log-space, using the empirical estimates provided in Table 1 as fixed input, and introduce a

prefactor $u$ as free parameter, which yields $u = 8.11$ (95% CI [7.69; 8.52]; Fig. 1d), i.e. the running speed predicted via Eq. (5) is about an order of magnitude too low. This discrepancy can be understood from first principles: The Hill- and the Borelli-number define a maximum speed increment that can result from a single contraction—but running animals can and do accelerate over multiple steps. During each step, some kinetic energy is lost to the collision of the leg with the ground[50], and some is actively removed by negative muscle work to bring the instantaneous vertical centre-of-mass velocity temporarily to zero; the combined loss consequently needs to be resupplied. A minimalist plausible assumption is that a constant fraction of the current CoM velocity is removed with each step, i.e. that the effective coefficient of restitution ($\eta$) is independent of animal size and speed. The acceleration profile is then asymptotic, in agreement with empirical observations, and the maximum speed is related to $\eta$ via $v_{\text{peak}} \propto v_i(1-\eta)^{-1}$ where $v_i$ is the maximum speed increment predicted by the relevant dynamic similarity index (see methods). We estimate $\eta \approx 0.885$ from available data (95% CI [0.879; 0.890]), which corresponds to $u = (1-\eta)^{-1} = 8.69$ (95% CI [8.26; 9.09]), in robust agreement with the fitted estimate. We conclude that the prediction of the maximum running speed enabled by the addition of $\eta$ to eq. (5) is in satisfactory qualitative and quantitative agreement with the empirically observed allometry of maximum running speed (see Fig. 1d).

## Discussion

Animals small and large move by doing work with muscle, but the absolute maximum running speed they can achieve varies non-monotonously with size[6]. In principal agreement with previous related work[14,18,19,28,36], we argue that this peculiar allometry arises because the limiting physical constraint changes with animal size; in distinction to this body of work, we submit that the two competing constraints are the kinetic energy and work capacity that characterise every musculoskeletal system. The competition between these two constraints is captured quantitatively by the physiological similarity index $\Gamma$[26], which thus emerges as a fundamental dimensionless number that characterises muscle-driven motion across scales: small animals move with small $\Gamma$ and can thus only run as fast as the kinetic energy capacity of their musculoskeletal system permits; large animals move with large $\Gamma$, and are consequently constrained by the work density of their muscle. In a sense, $\Gamma$ is to musculoskeletal dynamics what the Reynolds number, Re, is to fluid mechanics, a suggestion of analogy we now frame with three brief examples.

Much like an increase in the Reynolds number indicates a transition from a dominance of viscous to a dominance of inertial forces, an inspection of $\Gamma$ reveals a transition in the key physical ingredients that limit the dynamic mechanical output of muscle as animals get larger. Terrestrial scaling theory has traditionally focussed on the impact of the differential scaling of weight vs. muscle forces: Galileo famously speculated about size-specific variations in bone proportions[51], and Haldane painted a picture of a giant so threatened by the risk of bone fracture that they were unable to move[52]. The theory that the need for size-invariant bone stresses has driven size-specific adjustments to skeletal anatomy and animal posture has become textbook material[40,53]; many have speculated that the same need is to be held responsible for the reduction in absolute maximum running speed in the largest terrestrial animals [e.g.[10,15,16,54,55]]. The analysis presented here challenges this perspective, for it suggests that a possible stress limitation to speed, imposed by the differential scaling of muscle and weight force, $\kappa_g \propto m^{1/3}$, is second to the effect of the differential scaling of the characteristic maximal inertial force $F_{ci} = ma \propto mv_{max}^2\delta_{max}^{-1}$—the product of the body mass and a characteristic acceleration—and the maximal ground reaction force $F_{GRF}$, $\Gamma \propto F_{ci}F_{GRF}^{-1} \propto m^{2/3}$: inertial forces dominate muscle dynamics in large animals, but are of little relevance in small animals where accelerating muscle contractions become practically instantaneous. The competition between muscle

and gravitational force does eventually demand a decrease in speed, but this decrease does not have to reflect a stress-mitigation strategy per se; it can occur simply because the monotonous increase of $\kappa_g$ goes hand-in-hand with a change in the fraction of muscle work which flows into gravitational potential energy, $W_g W_m^{-1} = \kappa_g$ vs kinetic energy, $E_{kin} W_m^{-1} = 1 - \kappa_g$ [Fig. 1b, c and ref. 26]. For a large 15 g insect, the gravitational force consumes a mere 1% of every unit of muscle work, for a geometrically similar and moderately large reptile of 20 kg it demands about 10%, and only for a geometrically similar large mammal heavier than about 2 t does the majority of muscle work flow into gravitational potential energy (Fig. 1b, c and see Supplementary Fig. 1a). We conclude that the variation of the ratio between weight and muscle force is likely inconsequential for all but the heaviest animals, because the gravitational force is small compared to the maximum ground reaction force by virtue of necessity; the majority of terrestrial animals can perhaps be considered gravitationally indifferent[26,56–58].

A powerful application of the Reynolds number is the identification of conditions of dynamic similarity—experiments with very small or very large specimen can be impractical, but a suitable adjustment of the fluid viscosity enables accurate experiments with scaled models of more convenient size. In the same spirit, dimensional analyses of terrestrial locomotion may be used to predict characteristic gait parameters as a function of speed, to identify preferred speeds, to define characteristic speeds at which gait transitions occur, or even to estimate locomotor speed in extinct animals [e.g.5,22,39,59]. Indeed, meaningful comparison of locomotor performance across animals of different sizes is impossible without a similarity framework, and similarity arguments have consequently taken a central role in comparative work on animal locomotion. The dominant approach has relied on the Froude number, to the extent that dynamic similarity in terrestrial locomotion has become practically synonymous with equality of Froude numbers. It has been noted that the non-monotonous scaling of maximum running speed poses a threat to this line of argument [e.g.6,60]: preferred speeds and gait transition speeds tend to be proportional to the maximum sustained speed[3,39], but the Froude number predicts a monotonous scaling. However, alternative suggestions have remained absent, and the problem has remained unresolved. The identification of $\Gamma$ suggests a possible resolution of this conundrum: $\Gamma$ can be defined as the ratio of two dynamic similarity indices, the Hill- and the Borelli-number, $\Gamma = Bo^2 Hi^{-2}$ [26]: for $\Gamma << 1$, dynamically similar gaits involve equal Hill-numbers, for $\Gamma >> 1$, they occur at equal Borelli-numbers, and at intermediate values of $\Gamma \approx 1$, dynamically similar speeds occur at equal Hill-Borelli-numbers. These dynamic similarity conditions deviate meaningfully from the classic perspective painted by the Froude number, for the relevant index now varies with size. To illustrate the consequences of this shift in perspective, consider the classic work by Heglund and others, who demonstrated that quadrupeds with a body mass between 0.03 and 230 kg prefer to move with and change gaits at speeds that scale as $v \propto m^{0.18} – m^{0.22}$ [3,39]. This scaling is close to the prediction from the Froude number, $v_{Fr} \propto m^{1/6}$, but also close to the linearised scaling derived from the Hill-Borelli number in the same mass region, $v_{HiBo} \propto m^{0.18}$ (extracted for the trot-gallop transition, see Supplementary Note 2 and Supplementary Fig. 1b). However, the two indices yield strikingly different predictions when they are extrapolated to much smaller or much larger animals: for a small insect with a weight of 10 mg, dynamic similarity in terms of the Froude number predicts a characteristic speed $v_{Fr} = 0.26$ m s$^{-1}$, whereas the Hill-Borelli-number as defined by eq. (5) yields $v_{HiBo} = 0.04$ m s$^{-1}$—more than six times slower (see Supplementary Note 2 and Supplementary Fig. 1b). For a giant dinosaur of 40$t$, in turn, the Hill-Borelli-number suggests that no motion is possible without non-trivial musculoskeletal adaptations that break with geometric similarity, whereas the Froude number predicts neither a flattening nor a decrease in speed, and instead suggests that $v_{Fr} = 10.2$ m s$^{-1}$ (see Supplementary Note 2 and Supplementary Fig. 1b). These examples are perhaps a naive but by

no means an atypical application of the Froude number; an unconditional conflation of dynamic similarity with equality of Froude numbers is problematic, and may well yield equivalent speeds that are physiologically implausible, if not physically impossible[60,61]. We speculate that dynamic similarity in terrestrial locomotion may be better defined via equality of the Hill-Borelli-number, i.e. characteristic movements occur at equal fractions of the maximum possible performance—a hypothesis in need of further investigation.

Life at different Reynolds numbers promotes or even necessitates non-trivial adaptations in locomotor form and animal morphology. For example, jet propulsion becomes impractical at small Re due to excessive dissipation, and effective lift-based locomotion requires an Re larger than some critical intermediate value[62]. Similarly, life at different $\Gamma$ comes with altered demands on musculoskeletal 'design'. We briefly discuss two hypotheses on how this variation in physical demands may have resulted in non-trivial musculoskeletal adaptations in defiance of geometric similarity. First, we note that maximum speed is independent of muscle mass at small $\Gamma$. This prediction is noteworthy, for a variation in muscle mass alters the maximum net work and power the musculoskeletal system can deliver, and textbook scaling theory would consequently predict a concomitant decrease in maximum speed. But the maximum speed in the Hill-limit depends solely on the kinetic energy capacity, and is thus independent of maximal muscle work and power capacity. Selection on large muscles may thus be relaxed at small body sizes, but taxa with a lower muscle mass fraction $m_f$ are predicted to reach the Borelli-limit and the force limit at a lower critical mass. Consequently, such taxa should slow down at a lower body mass, and be limited to a smaller absolute size. Preliminary support for these hypotheses exists: $m_f$ is approximately geometrically similar within reptiles and mammals[63,64], but reptiles have a $m_f$ about a factor of two lower than mammals[64]. And indeed, reptiles run with approximately equal maximum speed as mammals at small body sizes (Fig. 1), but slow down at a lower body mass[16], and the largest extant reptile is about five times lighter than the largest extant mammal. A more quantitative test of this hypothesis is hampered by the absence of reliable maximum running speed measurements for reptiles larger than about 10 kg; of course, selection also does not act solely on maximum speed. Second, the functional significance of the gear ratio $G$ changes with size: for small $\Gamma$, a smaller $G$ maximises the kinetic energy capacity of the musculoskeletal system and thus the maximum speed. For large $\Gamma$, $G$ merely controls whether a unit of net work is done by a large force and a small displacement or vice versa; it consequently leaves the possible work output unaffected. However, large animals also move with large $\kappa_G$, and $G$ then also controls the partitioning of muscle work into $E_{pot}$ and $E_{kin}$ [26]. As a result, a larger gear ratio now allows larger speeds with equal muscle work output, in remarkable contrast to the canonical interpretation of $G$ as a parameter which controls force-velocity trade-offs. A deviation from geometrically similar gear ratios may also be beneficial for animals below the transition mass. Speed increases with size in this regime, demanding an equivalent increase in the body-mass-specific work output of muscle. How this increased work output is achieved is an open question, because it requires breaking with geometric similarity: a size-invariant gear ratio would assign all variation in work output to an increased muscle strain, which would then grow with substantial positive allometry. A small increase in the gear ratio reduces the maximum speed, in return for a substantial reduction of the necessary muscle strain. Small animals thus appear to benefit from small gear ratios, and large animals from large gear ratios—a pattern which resembles classic observations [we note that other non-mutually exclusive hypotheses on the benefit of scaling gear ratios exists, see e.g. refs. 40,41,65]. A robust assessment of the allometry of $G$, its mechanical effects, and identification of the functional demands which drive it, will have to await availability of a sufficiently large and phylogenetically diverse dataset, for the allometry of $G$ is confounded by

evolutionary history [see[41], and methods]. However, the conclusion that the largest extinct animals will have moved with larger gear ratios appears a physical necessity: for a muscle mass fraction of about 20% across all limbs, comparable to that in extant quadrupeds and bipeds[64], a gear ratio of $G = 0.3$ would result in static equilibrium at a body mass of about 15 t (assuming half of the limbs are in ground contact); this critical mass goes as $m_c \propto G^3$, and a gear ratio of $G = 1$ would thus suffice to achieve static equilibrium in a giant dinosaur of about 500 t, about 6–10 times heavier than existing estimates for the mass of the largest dinosaur[66].

The size-specific variation of $\Gamma$ and $\kappa_g$ has substantial consequences for the mechanical performance space accessible by muscle, and thus for 'optimal' musculoskeletal 'design' across animal size. Although the minimalist physical estimates presented in this work are rooted in first principles and show noticeable agreement with empirical observations, a mere consistency between a theory and the data it was constructed to explain ought not be mistaken for conclusive evidence–the available data carries uncertainty, simplifying assumptions and approximations introduce inaccuracies, the model includes many independent parameters, and alternative scaling arguments exist[11,14,18,19,28]. Extensive studies that integrate results from invertebrate and vertebrate taxa across a maximal size range and phylogenetic diversity are crucial to identifying adaptations to the challenges posed by the highlighted physical constraints; to so test theoretical prediction outside their original domain; to separate physical, developmental, and phylogenetic constraints; to advance our general understanding of the mechanical limits to terrestrial locomotion across animal size; and to ultimately explore the physical origin of similar patterns in the allometry of locomotor speed across running, flying and swimming animals[17,67]. The future of evolutionary biomechanics is bright.

## Materials

### Empirical data on maximum running speed

Data on maximum running speed was extracted from the literature[6,10,12,17,37,61,68–102], either directly from the text or tables where possible, or from figures, using Ankit Rohatgi's WebPlotDigitizer.v 4.6 (automeris.io/WebPlotDigitizer, $n = 632$). All data is available in a Source Data File. There is no way of knowing whether any measured speed represents a true maximum, and it can be reasonably argued that the only reliable estimates come from race horses, race dogs, and elite human sprinters. We submit that this uncertainty is sufficiently trumped by the variation introduced by the variation in body mass, which spans 11 orders of magnitude. This argument is in keeping with common practice in the literature: the general trend that a cheetah is faster than both a mouse and an elephant appears challenging to call into question. One further limitation requires commentary: the assembled data combines ontogenetic, static and evolutionary allometry into one single data set, i.e. it lumps data from individuals of the same species at different ontogenetic levels; of individuals of the same species at the same ontogenetic level but different size; and of individuals from different species. There are strong reasons to believe that ontogenetic, static and evolutionary allometries can and do differ [e.g. refs. 62,64,103–106], but the physical constraints that are the subject of this work are invariant to evolutionary biology, and apply equally well to each of these three levels. Although the relative magnitude of the involved physical quantities may well differ across phylogenetically distant groups, it is hard if not impossible to break with geometric similarity over many decades of body mass, so that the lumped analysis we conduct is plausible at least to first order.

### Empirical estimates for physical parameters of musculoskeletal systems

The physical analysis presented in the results has the advantage that it enables a direct quantitative estimation of the maximum running speed; it contains no empirical parameters void of physical meaning, and hence, in principle, does not require any statistical fitting routines. In order to provide a direct prediction, estimates for the relevant physical quantities were extracted from the literature (see Table 1), and we now briefly discuss critical aspects of this process.

First, and in general, the exact geometric similarity was assumed wherever available scaling exponents were consistent with or very close to this hypothesis (but see point three on the gear ratio). Geometric similarity is a parsimonious and plausible null hypothesis, and

**Table 1 | In order to directly predict the allometry of maximum running speed, estimates for the physical parameters that define the Hill- and the Borelli-limit were extracted from the literature: a representative fascicle length, $l_m$, the maximum muscle strain rate, $\dot{\varepsilon}_{max}$, the musculoskeletal gear ratio, $G$, the ratio between muscle and body mass, $m_f = m_m m^{-1}$, the muscle density, $\rho$, the maximum isometric muscle stress, $\sigma_{max}$, the effective coefficient of restitution, $\eta$, and the maximum strain, $\varepsilon_{max}$**

| | Parameter | Empirical estimate | Source | Comment |
|---|---|---|---|---|
| Primary parameters | $l_m$ | 0.03 m mass$^{1/3}$ kg$^{-1/3}$ | [64] | Averaged across hind- and forelimbs, and assuming geometric similarity. |
| | $\dot{\varepsilon}_{max}$ | 10 muscle lengths s$^{-1}$ | [111,112] | Faster strain rates have been reported, but likely represent extreme specialisation. |
| | $G$ | 0.3 | [41–43,107] | Arithmetic average; see text and Fig. 2. |
| | $m_f$ | 0.1 | [64] | Assuming geometric similarity, and that half of the total limb muscle mass contributes to acceleration during stance. |
| | $\rho$ | 1060 kg m$^{-3}$ | [113] | Density of muscle tissue. |
| | $\sigma_{max}$ | 250 kPa | [114–116] | |
| | $\eta$ | 0.89 | [61,109] | Effective coefficient of restitution. See text. |
| | $\varepsilon_{max}$ | 0.3 | | Yields a typical work density of 71 J kg$^{-1}$. |
| Derived parameters | $v_{Hi}$ | 1 m s$^{-1}$mass$^{1/3}$ kg$^{-1/3}$ | | Assuming geometric similarity. |
| | $v_{Bo}$ | 3.8 m s$^{-1}$ | | Assuming geometric similarity. |
| | $\Gamma$ | 0.07 mass$^{2/3}$ kg$^{-2/3}$ | | Assuming geometric similarity. |
| | $\kappa_g$ | 0.041 mass$^{1/3}$ kg$^{-1/3}$ | | Close to an upper bound estimated from experimental data in ref. [117], $\kappa_g \geq 0.05$ mass$^{1/3}$. |
| | $W_\rho$ | 71 J kg$^{-1}$ | | Consistent with typical estimates[114,116]. |

From these parameters, we derived first order predictions for the Hill- and Borelli-limit, $v_{Hi} = l_m \dot{\varepsilon}_{max} G^{-1}$ and $v_{Bo} = \sqrt{2\sigma_{max}\rho^{-1}m_f}$, the physiological similarity index, $\Gamma = v_{Hi}^2 v_{Bo}^{-2}$, the reduced parasitic energy, $\kappa_g = \rho g m_f^{-1}\sigma_{max}^{-1}l_m G^{-1}$, and the work density of muscle, $W_\rho = \sigma_{max}\varepsilon_{max}\rho^{-1}$. Estimates for $l_m$ and $m_f$ were obtained as arithmetic averages or via log-log regression from data on 31 mammalian and reptile species that varied across four orders of magnitude in body mass; the fascicle length here is the fascicle length of a hypothetical muscle which has a volume and physiological cross-sectional area equal to the sum of all relevant limb muscles[64]. All estimates assume that half of the total limb muscle mass contributes to the acceleration during stance.

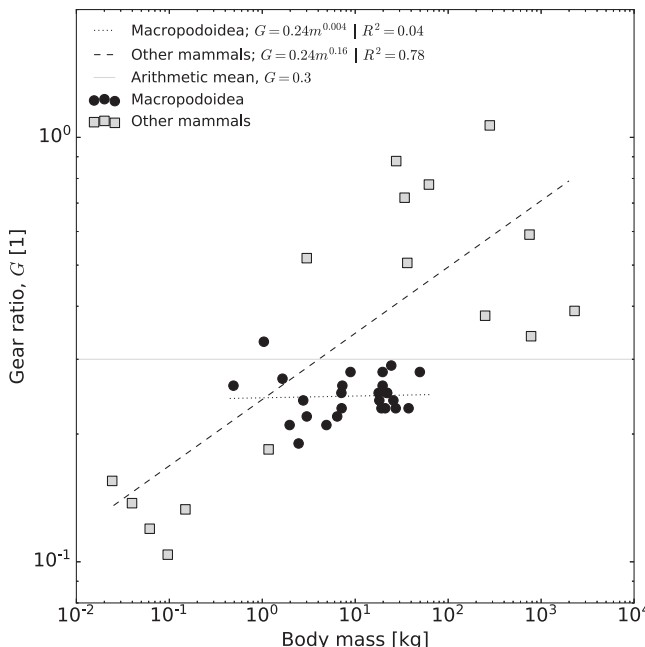

**Fig. 2 | The variation of the musculoskeletal gear ratio, G, with size across 42 vertebrate species varying by five orders of magnitude in mass[40–43,107].** In the initial across-clade analysis of the allometry of maximum running speed—which included invertebrates much smaller than 0.01 kg and vertebrates heavier than 2 t— an average gear ratio, $G = 0.3$ (solid line) was used, because it complies with the parsimonious assumption of geometric similarity, and because the gear ratio is confounded by evolutionary history, as evidenced by the different slopes for quadrupedal mammals vs. bipedal Macropodoidea, so that extrapolation bears significant risks [dashed vs. dotted line[41,107]]. The assumption of a size-invariant gear ratio is subsequently relaxed for the size range for which experimental data are available, and the consequences of a size-variable gear ratio are discussed. Source data are provided as a Source Data file.

thus the obvious starting point. Second, we note that the fascicle length, $l_m$ is an equivalent fascicle length, equal to the fascicle length of a hypothetical muscle which has a volume and physiological cross-sectional area equal to the sum of that of all relevant limb muscles, averaged across all limbs [for further detail, see ref. 64]. In practice, this definition reflects the physical reality that the speed of individual contractile elements linked in series is additive. The physiological cross-sectional area then follows in analogy, as the ratio between total limb muscle volume and equivalent fascicle length[64]. Third, we estimate an average gear ratio $G = 0.3 \pm 0.2$ from published data on 42 vertebrate species covering about five orders of magnitude in body mass [Fig. 2; data from[41–43,107]]. As is well established, the gear ratio increases significantly with size within quadrupedal mammals, $G \propto m^{0.16}$ [40,41, We note that this slope differs from the estimate reported in the original papers, as we included additional data.]. $G$ is however approximately size-invariant within the bipedal, hopping Macropodoidea [$G \propto m^{0.004}$;107]. We used a size-invariant gear ratio as the initial guess for our across-clade analysis to adhere to the parsimonious assumption of geometric similarity, and because the allometry of the gear ratio appears to be clearly confounded by evolutionary history, so that extrapolation to groups not represented in the available data is likely unreliable [Fig. 2 and[41]]. Further quantitative interpretation and in particular extrapolation to animals outside the size range for which data is available can be conducted once a larger and more phylogenetically diverse dataset is available. The functional significance of a systematic variation of $G$ with size, and its effect on maximum running speed, is critically evaluated in more detail in the results. Fourth and last, our analysis required estimation of an effective 'coefficient of restitution', $\eta$, which quantifies the speed loss

from collisions and unavoidable fluctuations in the vertical centre-of-mass velocity that accompany each step. We first briefly derive the relation between the speed increment per step ($v_i$), the effective coefficient of restitution ($\eta$), and the maximum possible speed ($v_{peak}$), and then estimate $\eta$ directly from empirical data. All statistical analysis below or elsewhere in the manuscript was conducted in R v. 4.3.2 pr Python 3.

## The effective coefficient of restitution

The Hill- and the Borelli-number provide upper bounds for the speed increment that can be achieved with a single contraction, but running animals accelerate over multiple steps. To estimate the absolute maximum running speed, an assumption is thus required on how these increments accumulate over a series of steps. If animals were able to add the same increment at every step without any intermediate loss of kinetic energy, the speed would simply be the product of the number of steps and the speed increment. However, every step is associated with collisional dissipation of energy by necessity[50], and with a further loss of kinetic energy because the vertical centre-of-mass velocity must be instantaneously zero at some point during stance. As a result, a fraction of the speed is lost with each step and needs to be re-supplied by muscle work. A parsimonious assumption is that this fraction, $1 - \eta$, is a constant independent of animal size and speed; initial empirical evidence in support of this assumption is provided below. A mathematical function can be constructed which encodes this assumption, and so describes the speed as a function of the step number, $n$, the loss factor, $1 - \eta$, and the maximum speed increment, $v_i$:

$$v = v_i \left[ \frac{1 - \eta^n}{1 - \eta} \right] \qquad (6)$$

This expressions predicts an asymptotic acceleration profile, which mirrors empirical data [e.g. refs. 69,108, and see Fig. 3a]. The asymptotic maximum speed follows as $v_{peak} = v_i (1 - \eta)^{-1}$, which is the prediction leaned on in the results.

In order to estimate $\eta$, we re-analysed data presented in refs. 61,109 for 9 species of birds and 13 species of lizards, varying between 8 g and 80 kg in body mass, and running at maximum speeds between 0.5 and 5.4 m s$^{-1}$. We selected sequences of multiple strides over which the average centre-of-mass (CoM) velocity remained approximately constant, to exclude acceleration and deceleration periods, and then extracted the maximal ratio of the minimum and maximum CoM velocity from these strides as an idealised upper bound for $\eta$. $\eta$ was then estimated as the slope of a zero-intercept ordinary least square regression of the minimum vs the maximum speed, which yielded $\eta = 0.89$ (95% CI [0.88, 0.92]), with a robust coefficient of determination $R^2 = 0.99$ (see Fig. 3b). Although encouraging, this result should be considered approximate. A more detailed analysis that relies on experiments designed for this analysis is required to derive firm conclusions about the variation of $\eta$ with phylogeny, animal size, and speed.

## Estimation of the transition mass between the Hill- and the Borelli-limit

Animals transition from the Hill- to the Borelli-limit when the speeds predicted by both are equal, i.e. when $\Gamma = v_{Hi}^2 v_{Bo}^{-2} = 1$. Using the estimates provided in Table 1 yields $\Gamma = 0.07$ m$^{2/3}$ kg$^{-2/3}$, which is unity for $m_T = (1/0.07)^{3/2} \approx 54$ kg.

## Estimation of the critical mass

The critical mass is the mass at which the maximal ground reaction force exactly balances the weight force, or in other words the mass for which $\kappa_g = 1$; the mechanical system is then in static equilibrium. Using the estimates provided in Table 1 yields $\kappa_g = 0.041$ m$^{1/3}$ kg$^{-1/3}$, so that $m_c = (1/0.041)^3 \approx 15$ t. This upper bound is close to the maximum body

**(a)**

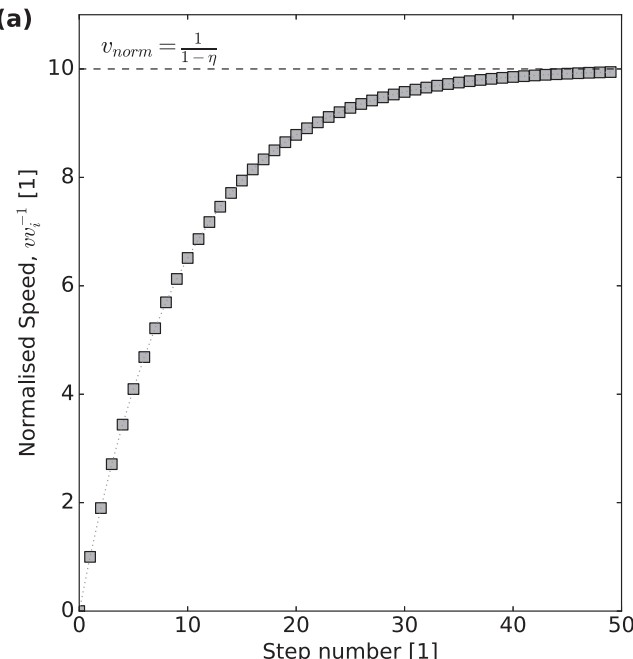

$$v_{norm} = \frac{1}{1-\eta}$$

**(b)**

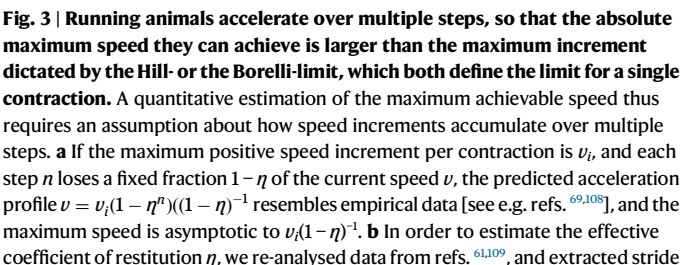

$v_{min} = 0.89 \, v_{max} \; [R^2 = 0.99]$

**Fig. 3 | Running animals accelerate over multiple steps, so that the absolute maximum speed they can achieve is larger than the maximum increment dictated by the Hill- or the Borelli-limit, which both define the limit for a single contraction.** A quantitative estimation of the maximum achievable speed thus requires an assumption about how speed increments accumulate over multiple steps. **a** If the maximum positive speed increment per contraction is $v_i$, and each step $n$ loses a fixed fraction $1 - \eta$ of the current speed $v$, the predicted acceleration profile $v = v_i(1 - \eta^n)((1 - \eta))^{-1}$ resembles empirical data [see e.g. refs. [69,108], and the maximum speed is asymptotic to $v_i(1 - \eta)^{-1}$. **b** In order to estimate the effective coefficient of restitution $\eta$, we re-analysed data from refs. [61,109], and extracted stride sequences over which the *average* speed remained approximately constant. Within these sequences, the maximum speed $v_{max}$ and minimum speed $v_{min}$ were extracted for each step, and the step with the maximal ratio $v_{min} v_{max}^{-1}$ was selected for further analysis, as it presents an upper bound for the minimal kinetic energy loss. Each data point in the plot represents a different individual ($n = 52$); the data represent experimental trials involving 9 species of birds and 13 species of lizards, varying between 8 g and 80 kg in body mass. $\eta$ was estimated from these data via an ordinary least-squares regression forced through the origin (hence the unusual shape of the confidence bands), which yielded $\eta = 0.89$ (95% CI [0.88, 0.92]). Source data are provided as a Source Data file.

mass estimate of 18 t for the largest extinct terrestrial mammals, proboscideans[110]. For a larger gear ratio of $G = 1$, $\kappa_g = 0.0125$, and $m_t \approx 500$ t; the critical mass is extremely sensitive to the gear ratio, and goes as its cube.

### Work partitioning into kinetic and gravitational potential energy

The reduced parasitic energy $\kappa_g = mg(F_{max} G)^{-1}$ can be directly related to the fraction of work which flows into gravitational potential vs kinetic energy, $W_g W_m^{-1} = \kappa_g$ vs $E_{kin} W_m^{-1} = 1 - \kappa_g$, respectively[26]. Using the estimates in Table 1 yields $\kappa_g = 0.041$ m$^{1/3}$ kg$^{-1/3}$, which for body masses of $m = 15$ g, 15 kg and 2 t yields $W_g W_m^{-1} = 1\%$, 10% and 52%, and accordingly $E_{kin} W_m^{-1} = 99\%$, 90% and 48%, respectively. The consumption of muscle work by gravitational potential energy remains below 10% for animals with a body mass below 10 kg, and the majority of terrestrial animals may thus be considered gravitationally indifferent (Supplementary Fig. 1a).

### Reporting summary

Further information on research design is available in the Nature Portfolio Reporting Summary linked to this article.

### Acknowledgements

This study was supported by an ARC Discovery Grant (DP180100220) to C.J.C. and D.L., and a Human Frontier Science Programme Young Investigator Award (RGY0073/2020) to D.L. This research has received funding from the European Research Council (ERC) under the European Union's Horizon 2020 research and innovation programme (grant agreement no. 851705, to D.L.).

### Author contributions

D.L.: developed theory; analysed data; made all figures; contributed toward conceptualisation, data, and interpretation; wrote the manuscript. P.J.B.: contributed toward conceptualisation, data, and interpretation; commented on manuscript drafts. T.J.M.D.: contributed toward conceptualisation, data, and interpretation; commented on manuscript drafts. C.J.C.: contributed toward conceptualisation, data, and interpretation; commented on manuscript drafts.

### Competing interests

The authors declare no competing interests.

### Data availability

All Source data are provided as a Source Data file. Source data are provided with this paper.

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
