## [Peer Review File · Nature Communications]

Dynamic similarity and the peculiar allometry of maximum running speedReviewers' Comments:

Reviewer #1:

Remarks to the Author:

I thoroughly enjoyed reading the article, which delves into predicting the running velocity of animals. The authors introduced the subject effectively and proposed a model grounded in Hill's muscle theory that correlates body mass with running speed. This approach yielded remarkable outcomes by condensing data spanning 10 orders of magnitude in mass into a singular curve. I found these results to be highly impressive, offering significant insights into locomotion.

However, for further refinement and to ensure its suitability for publication in Nature Communications, I suggest addressing the following points:

1. I observed that the manuscript's readability could be enhanced by reorganizing its structure. Redundant repetitions of results in various sections led to confusion. I recommend presenting the two asymptotics first, followed directly by the model, incorporating the gravitational effect while omitting Eq. 3 in favor of Eq. 5. Additionally, the introduction of the variable zeta seems extraneous given the abundance of mathematical quantities. The multiple discussions, in different places on the borelli limit of Hill limit could be consolidated into a single coherent analysis, thereby streamlining the manuscript.
 2. The discussion section appeared convoluted, with several repetitions of previously stated results. It might be beneficial to integrate most of the discussion's content directly into the results section for better context. Furthermore, certain aspects of the discussion seem more pertinent to the introduction, such as the principles of similarity (320->329). I recommend reconsidering the necessity of paragraphs 340-359.
 3. Given the existence of a recently published article (<https://www.nature.com/articles/s41467-023-41368-6>) that presents similar findings on swimming locomotion, drawing parallels between the two could strengthen the manuscript. Discussing the correlation between a ratio of L/L_c and Γ , and investigating potential transitions from endotherms to exotherms linked to changes in the scaling law, would add value.
 4. Considering the existence of an optimal stride frequency for human runners, it would be of incredible use if the authors could provide a predictive curve for running frequency as a function of body mass.
 5. As most of the theory in the manuscript has already been derived in a prior article published in PNAS 2023, there seems to be little necessity to rederive Eq. 5 in the Supplementary Information.
 6. It would be clearer for readers if Eq. 5 were presented using physical parameters instead of dimensionless numbers. A formulation like $v = v_{\text{reference}} \times \text{functon}(\Gamma, K_g)$ would enable readers to grasp the key takeaway more easily, especially when considering Fig. 1."
- I hope these revisions help to improve the clarity and structure of your text

Reviewer #2:

Remarks to the Author:

In this paper, the authors attempt to comprehensively address the question of the non-monotonic relationship between body mass and maximum running speed. This is a longstanding question in movement ecology and evolutionary biomechanics, and this paper goes far into exploring mechanistic and physically/physiologically realistic underpinnings of this scaling relationship. This is a significant step further than previous explorations of this phenomenon and is of interest to a wide audience. I appreciate the thorough and clear analysis put into this work, and the care taken to address comparisons to previous efforts along the same line of inquiry (e.g., in the SI).

I was excited to read this paper, and I think its broad reach and interdisciplinary nature makes it a good fit for the readership of Nature Communications. I only have some relatively minor comments that I would appreciate being addressed to improve the clarity and presentation of the paper.

Comments:

- It would be useful to perhaps have a table with all dimensionless numbers discussed throughout (in the main text vs in the supplement), with brief descriptions and corresponding expressions -- this would definitely facilitate understanding the relationships described in the results.
- Line 54: muscle is ubiquitous across (much of) the animals, but much of the tree of life (plants, fungi, protists, bacteria) don't use muscles for movement! It's completely fair that the authors hone in on muscles in this paper, which focuses on running. However, a slight reword here would be good to acknowledge the limits of muscles 'ubiquity'.
- Since it is used within, m_c should be explicitly defined in the caption of Figure 1 (to make the figure understandable in a self-contained fashion), not just in the text.
- Will the data used in Table 1 be made available for reproducibility purposes? At the moment, it was not present in the SI.

Reviewer #3:

Remarks to the Author:

This manuscript was a delight to read. I congratulate the authors on a well-reasoned and well-written argument. The beauty of proposing a scaling hypothesis is that any caveat or exception that I, or another reader, can devise can simply be framed as a necessary modification to mitigate the consequences of said hypothesis. The utility of the hypothesis is therefore determined by the extent to which it can be universally applied as a framework for studying diverse systems before the basic assumptions are violated.

With that in mind, I agree with the authors that an energy-based perspective of terrestrial locomotion is needed as more and more studies of comparative biomechanics integrate muscle physiology with whole-animal dynamics. Because terrestrial motion, by definition, is ruled by gravity, and (mostly) is powered by muscle, I expect that the proposed hypothesis will be a useful starting point for the study of diverse systems.

I have one comment related to the clarity of the manuscript:

I find the language at Line 306 that describes Γ as the ratio of maximum inertial force to GRF to be confusing (a reference to inertial forces is also included in the abstract). I believe the authors are using the term in reference to "effective inertia" introduced in Labonte 2023. I would argue that most authors are not familiar with this use of "inertia" and in the context of the paper are more likely to interpret this, as I first did, to be the "parasitic" force associated with the acceleration and deceleration of limbs during cyclical running. I believe clarification is needed here and perhaps throughout to introduce unfamiliar readers with the language from Labonte 2023.

We thank all reviewers for their time and constructive comments. The reviews are reproduced at verbatim below, alongside a point-by-point reply.

Reviewer #1 (Remarks to the Author):

I thoroughly enjoyed reading the article, which delves into predicting the running velocity of animals. The authors introduced the subject effectively and proposed a model grounded in Hill's muscle theory that correlates body mass with running speed. This approach yielded remarkable outcomes by condensing data spanning 10 orders of magnitude in mass into a singular curve. I found these results to be highly impressive, offering significant insights into locomotion.

Thanks kindly for the encouraging feedback.

However, for further refinement and to ensure its suitability for publication in Nature Communications, I suggest addressing the following points:

1. I observed that the manuscript's readability could be enhanced by reorganizing its structure. Redundant repetitions of results in various sections led to confusion. I recommend presenting the two asymptotics first, followed directly by the model, incorporating the gravitational effect while omitting Eq. 3 in favor of Eq. 5. The multiple discussions, in different places on the borelli limit of Hill limit could be consolidated into a single coherent analysis, thereby streamlining the manuscript.

We thank the reviewer for their thoughtful suggestion, which we gave careful consideration. After much discussion, we decided to leave the presentation as is, not because we do not find the reviewer's suggestion valuable and sensible, but because we believe that there are multiple sensible ways to present the argument, each with respective strengths and weaknesses. We believe the reviewer's suggestion would be most suitable for readers with a strong physical background who will not struggle with the equations and physical reasoning. However, a sizeable fraction of the likely readers of this paper will be biologists with less or even no physical and mathematical training, and we believe that making the manuscript as accessible to a wide an audience as possible is facilitated by a gentler, step-wise introduction--and indeed a little repetition--at little cost to the expert physicist reader. Because this is merely a question of style and personal preferences, it does not affect the scientific rigour or accuracy of this work.

Additionally, the introduction of the variable zeta seems extraneous given the abundance of mathematical quantities.

We removed ζ as suggested.

2. The discussion section appeared convoluted, with several repetitions of previously stated results. It might be beneficial to integrate most of the discussion's content directly into the results section for better context. Furthermore, certain aspects of the discussion seem more pertinent to the introduction, such as the principles of similarity (320->329). I recommend reconsidering the necessity of paragraphs 340-359.

We thank the referee for the careful reading of these sections of the work. We agree that a split into results and discussion often comes with some repetition, and tried to keep this to a reasonable minimum (we note that this split is recommended in Nat Comm's "Instruction to the authors", which we followed in the preparation of this article). The three main points made in the discussion, all related to a comparison of Γ with the Reynolds number, are novel, and distinct from the results presented in the earlier section. We consider some repetition useful, because it will help readers with a large variety of backgrounds to follow the progression of the argument more easily. A good example are the principles of similarity, which will be well familiar to some readers, but not to

others. We believe that the concrete example of how such similarity arguments are used in biology, presented in the lines referred to by the referee, will help readers less familiar with the concept to grasp it, even though similarity concepts are also mentioned in the introduction. Because this paper is purposefully written for a general audience, we consider increasing accessibility essential where reasonably possible.

After careful consideration of the reviewer's helpful suggestion, we decided to retain paragraphs 340-359; this is because the Froude number continues to dominate similarity analyses in research on terrestrial locomotion, and we think that this practise carries significant risks.

3. Given the existence of a recently published article (<https://www.nature.com/articles/s41467-023-41368-6>) that presents similar findings on swimming locomotion, drawing parallels between the two could strengthen the manuscript. Discussing the correlation between a ratio of L/L_c and F , and investigating potential transitions from endotherms to exotherms linked to changes in the scaling law, would add value.

When the excellent work by Rodriguez et al was published, our manuscript draft was practically complete. We now added a reference to this work at the end of the discussion, where we also express our excitement, shared with the reviewer, about the possibility for even broader comparative analyses, combining data on running, flying and swimming animals:

[...] to advance our general understanding of the mechanical limits to terrestrial locomotion across animal size; and to explore the physical origin of similarities in the allometry of locomotor speed across running, flying and swimming animals [17, 64].

We are currently working on a more expansive analysis, including a more detailed comparison to the work by Rodriguez et al. We decided not to comment on flight and swimming in detail here, to avoid falling into a similar trap as previous work by Hirt et al, which was arguably too hasty in announcing a “general law” (see discussion in the SI). We believe that a broad and robust comparison requires additional careful analyses and in particular new experimental data on the musculoskeletal parameters listed in Table S1 for swimming and flying animals. Instead of adding a discussion of these points in this manuscript, we prefer to give them the required space and careful analyses in separate follow-up work.

4. Considering the existence of an optimal stride frequency for human runners, it would be of incredible use if the authors could provide a predictive curve for running frequency as a function of body mass.

We entirely agree with the referee. Indeed, the variation of stride length and stride frequency across the phylogeny and body size spectrum of running animals continues to attract attention (see e.g. Granatosky, M. C., & McElroy, E. J. (2022). Stride frequency or length? A phylogenetic approach to understand how animals regulate locomotor speed. JEB 225: jeb243231), and presents a challenging, worthwhile and interesting open puzzle. We are currently in the process of assembling a sufficiently large comparative data set, including both vertebrate and invertebrate animals, to enable future physical analyses of this relationship in terrestrial locomotion. Assembly and analysis of this dataset is not sufficiently progressed to permit a fully rigorous treatment of this question yet, and so we refrain from addressing it in the present manuscript.

5. As most of the theory in the manuscript has already been derived in a prior article published in PNAS 2023, there seems to be little necessity to rederive Eq. 5 in the Supplementary Information.

Eq. 5 is a novel result, and was not derived in Labonte 2023. Eq. 3, however, has been derived previously, and the derivation is provided again in the SI, as this makes it easier to understand the derivation of Eq. 5, and renders the presentation more complete. We believe there is no harm in making it a little easier for the interested reader to find the relevant derivations, and to follow the key steps, and therefore left the complete derivation within the SI.

6. It would be clearer for readers if Eq. 5 were presented using physical parameters instead of dimensionless numbers. A formulation like $v = v_{\text{reference}} \times \text{functon}(T, K_g)$ would enable readers to grasp the key takeaway more easily, especially when considering Fig. 1."

Implemented as suggested (also for Eq. 3).

I hope these revisions help to improve the clarity and structure of your text

Reviewer #2 (Remarks to the Author):

In this paper, the authors attempt to comprehensively address the question of the non-monotonic relationship between body mass and maximum running speed. This is a longstanding question in movement ecology and evolutionary biomechanics, and this paper goes far into exploring mechanistic and physically/physiologically realistic underpinnings of this scaling relationship. This is a significant step further than previous explorations of this phenomenon and is of interest to a wide audience. I appreciate the thorough and clear analysis put into this work, and the care taken to address comparisons to previous efforts along the same line of inquiry (e.g., in the SI).

I was excited to read this paper, and I think its broad reach and interdisciplinary nature makes it a good fit for the readership of Nature Communications. I only have some relatively minor comments that I would appreciate being addressed to improve the clarity and presentation of the paper.

Thanks kindly for the encouraging feedback.

Comments:

- It would be useful to perhaps have a table with all dimensionless numbers discussed throughout (in the main text vs in the supplement), with brief descriptions and corresponding expressions -- this would definitely facilitate understanding the relationships described in the results.

Such a table is now provided in the SI (Table S2). We also reduced the number of dimensionless quantities used throughout the text, in response to a comment by reviewer one (ζ is no longer used).

- Line 54: muscle is ubiquitous across (much of) the animals, but much of the tree of life (plants, fungi, protists, bacteria) don't use muscles for movement! It's completely fair that the authors hone in on muscles in this paper, which focuses on running. However, a slight reword here would be good to acknowledge the limits of muscles 'ubiquity'.

Changed to: "the ubiquitous animal motor"

- Since it is used within, m_c should be explicitly defined in the caption of Figure 1 (to make the figure understandable in a self-contained fashion), not just in the text.

Implemented as suggested. We now also define the second parameter used within the figure (m_t). The relevant sections of the caption now read:

If the muscle force is independent of muscle strain rate, the transition between these regimes is sharp and occurs at a body mass m_t (see (c)); if muscle has force-velocity properties, it is more gradual (dot-dashed line). κ_g quantifies the fraction of muscle work which flows into kinetic vs gravitational potential energy. The energy demanded by gravity only becomes appreciable for large κ_g , eventually resulting in a sharp asymptote at a critical body mass, m_c , at which movement is no longer possible (grey dashed line, see (c)).

- Will the data used in Table 1 be made available for reproducibility purposes? At the moment, it was not

present in the SI.

Absolutely – the data will be made available without access restrictions (it has been uploaded to the SI with the revision).

Reviewer #3 (Remarks to the Author):

This manuscript was a delight to read. I congratulate the authors on a well-reasoned and well-written argument. The beauty of proposing a scaling hypothesis is that any caveat or exception that I, or another reader, can devise can simply be framed as a necessary modification to mitigate the consequences of said hypothesis. The utility of the hypothesis is therefore determined by the extent to which it can be universally applied as a framework for studying diverse systems before the basic assumptions are violated.

Thanks kindly for the encouraging feedback. We agree that scaling arguments immunise themselves against falsification to some extent. We were conscious of this risk, and our mitigation strategy has been to show that our argument does not only reproduce the observed variation of maximum running speed qualitatively, but also quantitatively, and across a very large range of body sizes and animal groups. Although we agree that the argument as presented is a scaling analysis in character, it gains further substantial support by this quantitative agreement.

With that in mind, I agree with the authors that an energy-based perspective of terrestrial locomotion is needed as more and more studies of comparative biomechanics integrate muscle physiology with whole-animal dynamics. Because terrestrial motion, by definition, is ruled by gravity, and (mostly) is powered by muscle, I expect that the proposed hypothesis will be a useful starting point for the study of diverse systems.

I have one comment related to the clarity of the manuscript:

I find the language at Line 306 that describes Gamma as the ratio of maximum inertial force to GRF to be confusing (a reference to inertial forces is also included in the abstract). I believe the authors are using the term in reference to “effective inertia” introduced in Labonte 2023. I would argue that most authors are not familiar with this use of “inertia” and in the context of the paper are more likely to interpret this, as I first did, to be the “parasitic” force associated with the acceleration and deceleration of limbs during cyclical running. I believe clarification is needed here and perhaps throughout to introduce unfamiliar readers with the language from Labonte 2023.

Thanks for pointing out this potential source of confusion. We indeed mean an inertial force, and not the effective inertia; we agree that this point is not obvious. It emerges from a general characteristic of dimensionless numbers: they can be interpreted in a number of ways, in particular when they involve more than two physical quantities. Γ , the dimensional quantity referred to as “effective inertia” in Labonte 2023, can be interpreted as the ratio of two characteristic speeds, energies, displacements, times, or – as in the line in question – of two forces. One of these forces, the ground reaction force, is intuitive; what remains is the mass multiplied with a term which must have the same dimension as an acceleration, so leading to $\Gamma = m a / F_{GRF}$. The term ($m a$) is what we refer to as a characteristic inertial force. We believe that this interpretation is useful to illustrate (i) the analogy to the Reynolds number, often introduced as the ratio of inertial to viscous forces (though it could also be interpreted as the ratio between kinetic and viscous dissipative energy), and (ii) the contrast with the common assumption that the key force ratio is that of the gravitational to the ground reaction force. To alert the reader of this subtlety, we now provide an in-line qualitative definition alongside the mathematical expression:

[...] is second to the effect of the differential scaling of the characteristic maximal inertial force [...]---the product of the body mass and a characteristic acceleration---and the maximal ground

| reaction force [...] (Line 306-307)

Reviewers' Comments:

Reviewer #1:

None

Reviewer #2:

Remarks to the Author:

I remain enthusiastic about the premise of this paper and am satisfied with the careful and thoughtful response of the authors to all my prior (minor) concerns. I am now happy to recommend acceptance of this manuscript for publication to Nature Communications, and look forward to seeing it out.

Reviewer #3:

Remarks to the Author:

I'm satisfied with the responses to my previous comments and request no further changes at this time.